# Spectrum of Terahertz Emission from Single-Color Filament Plasma under Different Laser Beam Focusing

Georgy Rizaev [1], Dmitrii Pushkarev [1] and Leonid Seleznev [1,2,*]

[1] P. N. Lebedev Physical Institute of the Russian Academy of Sciences, Moscow 119991, Russia
[2] Faculty of Physics, Lomonosov Moscow State University, Moscow 119234, Russia
[*] Correspondence: seleznev@lebedev.ru

**Abstract:** The spectrum of terahertz radiation generated in plasma of a single-color laser filament is observed, using a new technique based on obtaining two-dimensional angular distributions at different frequencies. It is shown that the maximum of the spectrum occurs in the low-frequency region for different laser pump focusing conditions. It is demonstrated that with the initial beam numerical aperture growth the generation of terahertz radiation at high frequencies increases more intensely compared to low frequencies.

**Keywords:** filamentation; terahertz radiation; plasma channel

## 1. Introduction

The radiation of the terahertz spectral range (0.1–10 THz) attracts attention because of the variety of possible applications [1–3]. One of the sources of such emissions is laser-induced gas plasma [4], including that created during the filamentation of femtosecond pulses [5]. Initially, schemes for generating terahertz radiation in filament plasma assumed the presence of an external electrostatic field [6,7] or second harmonic pulses (the two-color scheme) [8,9]. But, after the reports of first transverse [10,11] and then forward conical terahertz emission [12,13] in the process of single-color filamentation, this method was considered attractive for its simplicity, despite the relatively low energy output [14]. The spectral composition of such a source is spatially inhomogeneous and depends on the propagation angle: the lower the radiation frequency, the larger the angle is to the axis at which it propagates [12,15,16]. At the same time, when a paper regarded an experimentally measured terahertz spectrum, it was collected from a limited angle [17–19]. However, as was predicted in [20] and experimentally demonstrated in [21], in some cases significant terahertz signals from a short filament can appear even at angles larger than 90°, including cases of single-color filamentation [22]. The radiation propagating at these angles is almost impossible to be collected by optical methods. Thus, it is evident that in the spectra from [17–19] a large part of the low-frequency radiation that propagated at large angles was not taken into account. The aim of the present work was to develop and apply a technique for evaluating the spectrum of terahertz radiation from a single-color filament plasma, and to study the dependence of that spectrum on the the initial laser beam focusing conditions. The technique is based on obtaining full two-dimensional spatial distributions of the terahertz radiation at different frequencies, including propagating at significant angles to the optical axis.

## 2. Materials and Methods

The experiment was conducted with the pulses from the titan-sapphire laser system (Avesta Ltd., Moscow, Russia ) with a central wavelength of 740 nm and a duration of 90 fs. The beam diameter was 8 mm at $1/e$ level. The pulse energy in the experiment was 3 mJ. The polarization of the laser beam was horizontal. For providing external geometric focusing, we used spherical mirrors with focal distances of 0.25 m and 1 m. For the purpose

of increasing the difference in numerical apertures, for milder focusing, we additionally telescoped the laser beam, so that its diameter decreased to 3 mm. The resulting numerical apertures, defined as the ratio of the initial beam radius at level $1/e$ to the mirror focal length, were 0.024 and 0.002. The registration of the terahertz radiation was carried out with a superconducting bolometer (Scontel, Moscow, Russia), which had two receiving channels: the first one had a sensitive detector element based on MoRe and a sensitivity range of 0.3–12 THz; the second had a detector based on NbN and an operating range of 0.1–6 THz. In order to measure the distribution of the terahertz radiation on the horizontal plane, the bolometer was placed on a movable rail, which could be rotated around the vertical axis passing through the spherical mirror focal point (Figure 1a). To make measurements at different vertical angles, the spherical mirror was set on a vertical stand and could be moved within 12 cm up or down from its position at the level of the bolometer input window (Figure 1b). For each vertical position of the mirror, the beam axis was realigned, so that the mirror focus returned to its original point on the plane, perpendicular to the optical axis. This technique made it possible to rotate the beam axis around the geometric focal point, which was considered the source of the terahertz radiation. As a result, the terahertz radiation propagating at the corresponding vertical angle was directed to the bolometer input window. The bolometer window had a size of 10 mm and was located at a distance of 40 cm from the source, which corresponded to the angular resolution of the scheme, of about 1°. Taking into account the scheme resolution, the horizontal displacement of the focal point during the realignment could be neglected, as the vertical shift of the mirror in all cases was noticeably less than its focal length. In this scheme, the zero value of the vertical and horizontal angle corresponded to the conditions when the laser beam was directed into the bolometer input window. We also picked out separate spectral components of terahertz radiation by inserting in front of the bolometer input window one of the bandpass filters, the transmission functions of which are given in Figure 1c.

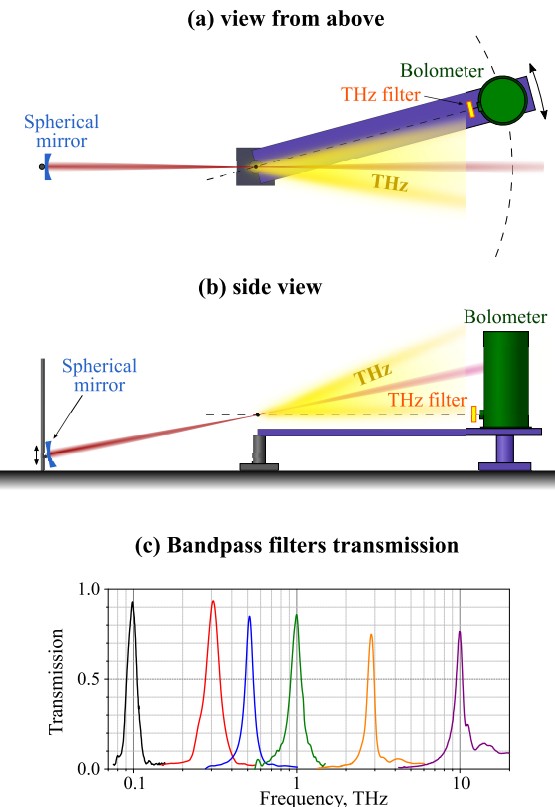

**Figure 1.** Experimental setup: (**a**) view from above; (**b**) side view; (**c**) transmission of the bandpass filters used in the experiment.

## 3. Results

Examples of the experimentally obtained two-dimensional angular distributions of terahertz radiation are given in Figure 2. The zero angle corresponded to the initial laser beam propagation axis. With increasing frequency, the propagation angles of the terahertz radiation decreased. At frequencies of 0.3 THz and 10 THz, the distribution profile had a shape close to a ring shape. A slightly different picture was observed at a frequency of 3 THz, where the distribution of the terahertz radiation had two maxima located on an axis perpendicular to the polarization of the laser radiation. It is worth noting that a similar pattern had been observed earlier in [18,23–25]. If the pattern of the terahertz radiation is axisymmetric—as, for example, in the case of DC-biased filaments—its spectrum can be reconstructed by integrating terahertz signals from one-dimensional angular distributions, as was done in [26]. But, as the obtained two-dimensional distributions of the terahertz emission from a single-color filament, especially at the frequency of 3 THz, does not have axial symmetry, any given one-dimensional angular profile cannot be considered as representative, to make conclusions on its contribution to the spectrum.

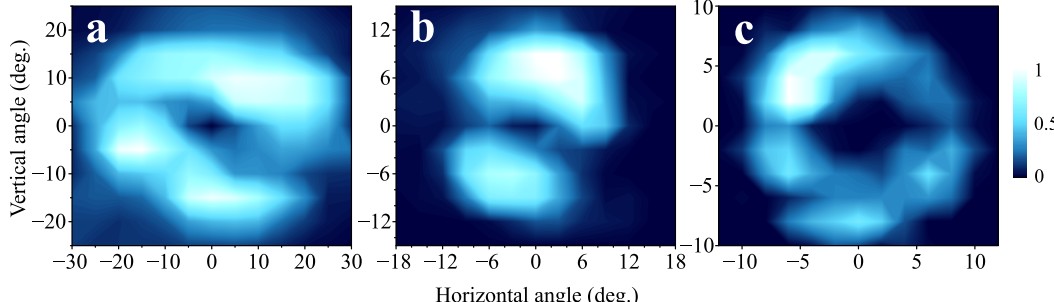

**Figure 2.** Two-dimensional distributions of the terahertz radiation at frequencies of (**a**) 0.3 THz, (**b**) 3 THz and (**c**) 10 THz. The numerical aperture of the laser beam was 0.024.

To estimate the spectrum of the terahertz emission, we analyzed the angular distributions measured in the same conditions with all available bandpass filters (from 0.1 to 10 THz). Importantly, this technique allowed us to take into account the radiation propagating at appreciable angles to the optical axis. For example, in our experiments for a frequency of 0.1 THz and a numerical aperture of 0.024, terahertz radiation was observed at angles up to 60° from the optical axis of the laser beam. After obtaining the full two-dimensional pattern of terahertz radiation at each frequency, the resulting signals were integrated over the entire spatial distribution. Having taken into account the spectral sensitivity of the bolometer and the filters transmission functions, we recalculated the obtained data, to ascertain the spectral intensity in arbitrary units.

The resulting spectra of the terahertz emissions for the two numerical apertures of the laser beam are shown in Figure 3. For both focusing conditions, the spectrum had a maximum in the low-frequency region. With increasing terahertz frequency, the spectral intensity of the radiation decreased monotonically. In the case of tighter focusing (NA = 0.024) for frequencies less than 1 THz, the spectral intensity was approximately one order of magnitude higher than for the milder focusing (NA = 0.002). At a 3 THz frequency, the difference in amplitudes was more than two orders of magnitude, and the signal at 10 THz for a beam numerical aperture of 0.002 was not recognizable in the experiments. Thus, as the numerical aperture of the laser beam increased, the amplitude of the terahertz radiation grew, and the enhancement of the generation of higher terahertz frequencies was more pronounced compared to the lower ones.

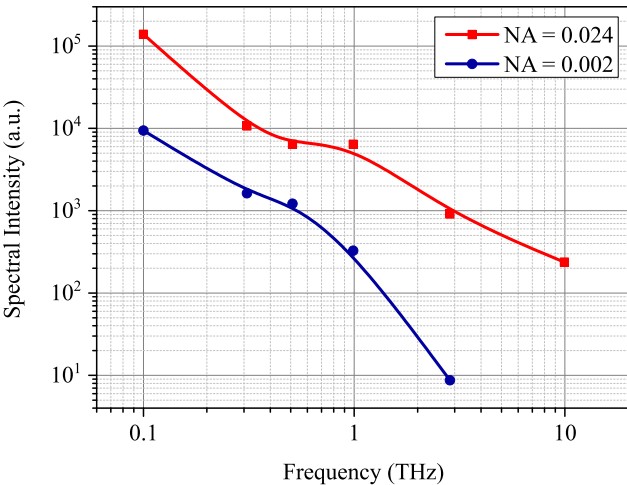

**Figure 3.** Spectrum of the terahertz emission at two different numerical apertures of the laser beam.

## 4. Discussion

The spectra obtained in the present study, however, differ significantly from the ones observed in [17–19,24], where the maximum of the terahertz radiation spectrum was located at frequencies larger than 0.5 THz. Furthermore, in [19] the frequency of the spectrum maximum increased with tightening of the focusing and could even exceed 2 THz. One of the reasons for such discrepancy might be the different angle range from which the terahertz radiation was collected. For more clarity, we additionally show angular distributions of terahertz radiation on the vertical plane for two numerical apertures, NA = 0.01 and NA = 0.05, in Figure 4. The choice of the numerical apertures was governed by the following consideration: the first one (NA = 0.01) corresponded to the focusing conditions in the work of [17] and was close to the focusing conditions with a lens $f = 35$ cm from the work of [19], while the second one was close to the focusing conditions with a lens $f = 7$ cm in the work of [19].

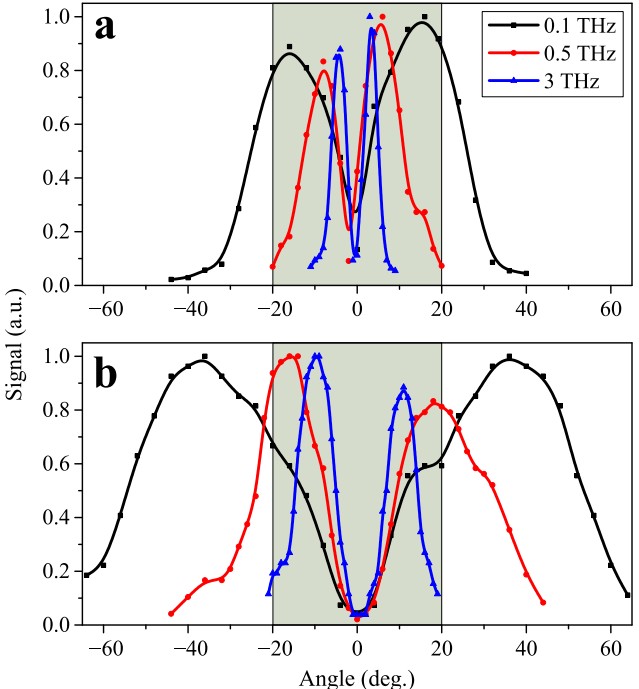

**Figure 4.** Normalized angular distributions of terahertz radiation at individual frequencies for different numerical apertures of: (**a**) NA = 0.01; (**b**) NA = 0.05.

As can be seen in Figure 4, by focusing the beam more tightly, the propagation angles of the terahertz radiation increased significantly. Let us assume that terahertz radiation is collected from a limited angle of, for example, 20° from the optical axis—the corresponding area in Figure 4 is highlighted with background color. For looser focusing (Figure 4a), the distributions of 0.5 and 3 THz fully lie inside the collecting area, and for a frequency of 0.1 THz a significant part of the radiation is taken into account. Meanwhile, for tighter focusing (Figure 4b), only a 3 THz pattern fits into the designated angle, the frequency of 0.5 THz partially falls into the selected area, and the main part of the radiation with a frequency of 0.1 THz remains outside the detection area. This explains the increase in the maximal frequency of the terahertz radiation spectrum at tighter focusing conditions when it is collected from a limited angle from the axis. Taking into consideration that in paper [19] the collection angle was about 25° from the axis, while in [17] it was about 14°, there are no essential contradictions with the spectra from the current work.

It is worth noting that the spectrum of terahertz radiation from a single-color filament plasma is spatially inhomogeneous. That is why from a potential applications point of view, instead of looking at the overall spectrum it is essential to know the angular distribution of the different spectral components. Nevertheless, our estimations provide information about the possibility of selecting certain spectral components and the share they have in an overall spectrum compared to other frequencies.

## 5. Conclusions

In conclusion, in our work we applied a new technique for estimating the spectrum of terahertz radiation generated in the process of single-color filamentation, which makes it possible to take into account, among other things, radiation propagating at significant angles to the axis. We showed that the major part of the integral energy of terahertz emission lies in a low-frequency region and that the maximum of the spectra has a frequency of about 0.1 THz or below. In the case of high numerical apertures of the laser beam, low-frequency terahertz radiation propagates at large angles to the axis; therefore, the limitation of the collection angle of terahertz radiation significantly determines the observed spectrum. We also experimentally demonstrated that as the focusing of the initial laser pulse becomes tighter, the intensity of the terahertz radiation increases over the entire frequency range, and that this growth is more pronounced in the high-frequency region.

**Author Contributions:** Conceptualization, methodology, investigation and data curation, G.R., D.P. and L.S.; writing—original draft preparation, G.R.; writing—review and editing, D.P. and L.S.; visualization, G.R.; supervision, project administration and funding acquisition, L.S. All authors have read and agreed to the published version of the manuscript.

**Funding:** This work was supported by the Russian Science Foundation (21-49-00023) and by the National Natural Science Foundation of China (12061131010).

**Institutional Review Board Statement:** Not applicable

**Informed Consent Statement:** Not applicable.

**Data Availability Statement:** The data underlying the results presented in this paper are not publicly available at this time but may be obtained from the authors upon reasonable request.

**Acknowledgments:** The authors would like to thank Daria Mokrousova for fruitful discussions and assistance in preparing the manuscript.

**Conflicts of Interest:** The authors declare no conflict of interest.

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
