# Peer review of "Spectrum of Terahertz Emission from Single-Color Filament Plasma under Different Laser Beam Focusing"

_photonics, doi:10.3390/photonics10101161_

Round 1
Reviewer 1 Report
· Please show on figure 1(a) and (b) where the vertical angle and the horizontal angle are zero;
· Line 39 page 2, can you explain more here ‘In the vertical plane, the angles were varied by moving and realigning the focusing mirror, so that the beam axis was correspondingly rotated around the focal point (Fig. 1b) '?
Figure 1(b), how high the spherical mirror can go? Not sure how you can use this mirror to scan the THz profile and keep the focus not moving?
· When calculating the numerical apertures 0.024 and 0.002 from the focal lengths 0.25 and 1 m, how do you measure the beam radius of 740 nm laser focus?
To fully understand the difference between your results with the previous publications, the profiles of plasma channel luminescence are needed, the numerical apertures alone are not sufficient.
· In Figure 2, Is THz not generated in the centre of the ‘donut’ shape profile, or is this due to the high power 740 mm laser that led the THz undetectable there?
Reviewer 2 Report
The manuscript of Rizaev et al. is devoted to the experimental estimation of the spectrum of THz emission from single-color filament. Authors measured the two-dimensional patterns of THz emission at the selected frequencies from 0.1 to 10 THz determined by the bandpass filters used. Then they integrated them taking into account the sensitivity of the bolometer and the transmittance of the bandpass filters. As a result of this routine, Authors reconstructed the spectral power at the central frequencies of the filters. The reconstructed spectra have the maxima below 0.1 THz, i.e. at the much lower frequencies as compared to earlier studies. The results are interesting and could be published in Photonics after Authors improve the following minor comments:
1. Indicate the beam diameter before and after telescope (lines 29–33).
2. The significant THz signal from a very short filament can appear even at the angles larger than 90 degrees. It was predicted theoretically in [Opt. Lett. 36, 3166 (2011)] and observed for the more powerful THz emission from two-color filament in [Appl. Phys. Lett. 114, 081102 (2019)]. I recommend to cite these papers (lines 15–17).
3. The routine of the spectrum reconstruction through the integration of THz angular distribution was used by Authors in the case of axial symmetry [Opt. Lett. 46, 5497 (2021)]. I recommend to cite this paper (lines 62–63).
